# Feasibility of a patient-oriented navigation programme for patients with lung cancer or stroke in Germany: Protocol of the CoreNAVI study

Kathrin Gödde[1]*, Hella Fügemann[2], Ute Goerling[3], Ulrike Grittner[4], Raphael Kohl[5], Andreas Meisel[6], Thomas Reinhold[7], Susanne Schnitzer[5], P. Markus Deckert[8], Nikolaj Frost[9], Stephan J. Schreiber[10], Nina Rieckmann[1], Christine Holmberg[2,11]

1 Institute of Public Health, Charité—Universitätsmedizin Berlin, Freie Universität Berlin and Humboldt-Universität zu Berlin, Berlin, Germany, 2 Institute of Social Medicine and Epidemiology, Brandenburg Medical School Theodor Fontane, Brandenburg an der Havel, Berlin, Germany, 3 Charité Comprehensive Cancer Center, Charité—Universitätsmedizin Berlin, Freie Universität Berlin and Humboldt-Universität zu Berlin, Berlin, Germany, 4 Institute of Biometry and Clinical Epidemiology, Charité—Universitätsmedizin Berlin, Freie Universität Berlin and Humboldt-Universität zu Berlin, Berlin, Germany, 5 Institute of Medical Sociology and Rehabilitation Science, Charité—Universitätsmedizin Berlin, Freie Universität Berlin and Humboldt-Universität zu Berlin, Berlin, Germany, 6 Center for Stroke Research (CSB), Charité—Universitätsmedizin Berlin, Freie Universität Berlin and Humboldt-Universität zu Berlin, Berlin, Germany, 7 Institute of Social Medicine, Epidemiology and Health Economics, Charité—Universitätsmedizin Berlin, Freie Universität Berlin and Humboldt-Universität zu Berlin, Berlin, Germany, 8 Oncology and Palliative Care and Faculty of Health Sciences, Department of Hematology, Both: Brandenburg Medical School Theodor Fontane, Brandenburg an der Havel, Berlin, Germany, 9 Department of Infectious Diseases and Pulmonary Medicine, Charité—Universitätsmedizin Berlin, Freie Universität Berlin and Humboldt-Universität Zu Berlin, Berlin, Germany, 10 Department of Neurology, Oberhavel Kliniken, Clinic Hennigsdorf, Hennigsdorf, Germany, 11 Faculty of Health Sciences, Brandenburg Medical School Theodor Fontane, Neuruppin, Germany

☉ These authors contributed equally to this work.
* kathrin.goedde@charite.de

## Abstract

### Background

Patient navigation programmes were introduced in the United States and recently gained interest in Germany, where the health care system is fragmented. Navigation programmes aim to decrease barriers to care for patients with age-associated diseases and complex care paths. Here we describe a feasibility study to evaluate a patient-oriented navigation model that was developed in a first project phase by integrating data about barriers to care, vulnerable patient populations and existing support services.

### Methods

We designed a mixed-methods feasibility study that consists of two two-arm randomized controlled trials aligned with observational cohorts. The intervention group of the RCTs gets support by personal navigators for 12 months. The control group receives a brochure with regional support offers for patients and caregivers. The feasibility of the patient-oriented navigation model for two prototypic age-associated diseases, lung cancer and stroke, is

**Data Availability Statement:** No datasets were generated or analysed during the current study. All

relevant data from this study will be made available upon study completion.

**Funding:** CoreNAVI is part of the research consortium "NAVICARE – Patient-oriented health services research" and is funded by the German Ministry of Education and Research (01GY1911). The funders had and will not have a role in study design, data collection and analysis, decision to publish, or preparation of the manuscript.

**Competing interests:** The authors have declared that no competing interests exist.

evaluated with regard to its acceptance, demand, practicality and efficacy. This investigation includes process evaluation measures with detailed documentation of the screening and recruitment process, questionnaires about satisfaction with navigation, observant participation and qualitative interviews. Estimates of efficacy for patient-reported outcomes are obtained at three follow-up time points including satisfaction with care and health-related quality of life. Furthermore, we analyze health insurance data from patients of the RCT insured at a large German health insurance (AOK Nordost) to investigate heath care utilization, costs and cost effectiveness.

## Trial registration

The study is registered at the German Clinical Trial Register (DRKS-ID: DRKS00025476).

## Introduction

Patient navigation (PN) was first developed in the USA to reduce health disparities in cancer care [1–3]. Person navigators of varying professional background support patients with complex conditions and a variety of chronic diseases or their family caregivers in overcoming barriers in healthcare. However, concepts differ in navigator roles and tasks [4]. Reviews on PN programmes in cancer care suggest that such programmes reduced time to treatment initiation and adherence in vulnerable patient groups in the US health care system [5,6]. However, data on their effectiveness regarding other healthcare indicators and patient outcomes is limited. In addition, little is known about the feasibility of PN programmes in different health care settings and about the success of their outreach strategies in order to include patients most in need of support. Factors of feasibility and effectiveness may vary considerably depending on the health care system it is applied to. The German health care system is fragmented with insufficient professional coordination for complex health care situations, such as chronic illness or multi-morbidity of patients. It is separated in inpatient and outpatient care with separate reimbursement rules, meaning that continuity of care between in-hospital and ambulatory providers is often not ensured [7–10]. Similarly, ambulatory care is not centred on general practitioners as gatekeepers and social care approaches are not regularly available outside the hospital. Thus, care coordination is mostly left to patients and / or their caregivers. Accordingly, studies have shown the importance of social networks of patients for good patient care in Germany [11,12]. Additional care models are needed in light of the demographic changes of an increasing elderly population with expected increasing prevalence of age-associated diseases and multi-morbidity and decreasing informal social networks for the older population [13].

To develop a navigation programme that is patient-oriented [14] and may adapt to the German context, we previously conducted a mixed-methods study including secondary data analyses [15,16], qualitative investigation of patients' needs, and expert interviews [17,18]. We integrated results from these analyses and developed a PN model to fit the care needs of stroke and lung cancer patients [19]. Using this multi-perspective, empirically based approach, we developed a patient-oriented PN programme for patients with age-associated, chronic diseases. It aims to support patients to receive optimal care addressing their individual goals and preferences in a first project phase.

Here, we present the study protocol of the CoreNAVI Study, a feasibility study to evaluate this PN programme designed to improve patient-oriented health care delivery for patients

with age-associated diseases, stroke or lung cancer, in ambulatory care in Germany. Core-NAVI is a sub-project of the NAVICARE network for patient-oriented health services research, which aims to reduce barriers and inequalities in the care of patients with age-associated diseases.

## Material and methods

### Aims and objectives

The aim of the study is to investigate the feasibility of a patient-oriented navigation programme in the German health care context. We chose to test this in two patient groups: persons with stroke and persons with lung cancer. We further aim to provide estimates of efficacy regarding selected patient reported outcomes, health care utilization and costs. The following main research questions will be addressed:

- Is the patient-oriented, individualized navigation programme that was developed in the first CoreNAVI project phase feasible (as indicated by acceptance, demand and practicality)?

- Does a patient-oriented navigation programme impact estimates of efficacy (patient reported outcome: satisfaction with care)?

  Additional secondary research questions are:

- Do we reach the patients that we assume most vulnerable (i.e. at risk of not receiving healthcare according to their preferences, values and needs) with the planned strategy?

- Which factors in the delivery process are associated with the (success or failure in the) acceptance, demand and practicality (feasibility) of the intervention?

- Is the timing, setting and mode of contact of the initial recruitment and navigation strategy optimal or should alternatives be considered?

- Which factors moderate the feasibility and the efficacy of the intervention?

### Study design

We evaluate the PN programme via a mixed-methods study with separate two-arm randomized controlled trials (RCTs) aligned with observational cohorts for stroke and lung cancer (one for each index disease; see Figs 1 and 2). This is an open-labelled study since blinding was not possible. Patient-reported outcomes and healthcare indicators are assessed at multiple time points during the follow-up (see Fig 2). Along the RCTs a process evaluation is performed including detailed documentation of screening, recruitment and intervention processes, a qualitative study including participant observation and personnel and participant interviews as well as an economic evaluation of the programme within a subgroup of patients in the RCT insured with a large German health insurance (Fig 2).

### Study registration

The study is registered at the German Clinical Trial Register (DRKS-ID: DRKS00025476). Registration was on 04.06.2021 (Version 1).

### Study setting

Active recruitment sites of patients are inpatient (stroke units) or specialized outpatient clinics for lung cancer in Berlin and Brandenburg, Germany. In addition, a range of rehabilitation

| | | | | STUDY PERIOD | | | | |
|---|---|---|---|---|---|---|---|---|
| | Enrolment | Baseline | Allocation | Allocation information | Follow-Up | | | |
| TIMEPOINT** | -t₁ | t₀ | | | | t₁ | t₂ | t₃ |
| ENROLMENT: | | | | | | | | |
| Eligibility screen | X | | | | | | | |
| Informed consent | X | | | | | | | |
| Allocation | | | X | | | | | |
| Allocation information | | | | X | | | | |
| INTERVENTIONS: | | | | | | | | |
| Arm 1: Navigation | X | X | X | X | | •————————• | | |
| Arm 2: Control | X | X | X | X | | | | |
| Cohort | X | X | | | | | | |
| ASSESSMENTS: | For detailed list of outcome variables see tables 1 - 4 | | | | | | | |
| | | X | | | | X | X | X |

**Fig 1. Schedule for the enrolment, intervention and data collection according to SPIRIT checklist [20].**

clinics in the Berlin and Brandenburg region have been selected at which study materials, including flyers and posters advertising the study, are distributed to patients.

The navigation intervention begins after in-patient care is completed. A navigator is assigned to a patient as constant contact for one year including personal meetings, home-visits, phone calls, digital visits and email conversation.

## Participants

Inclusion Criteria are:

- Patients with confirmed diagnosis of stroke/TIA (ICD-10 codes: G45.x, I60.x, I61.x, I63.x, I64.x, H34.x (since 01.08.2021), H47.0 (since 01.05.2022) or confirmed diagnosis of lung cancer (ICD-10 codes: C34.1, C34.2, C34.3, C34.8, C34.9, C97)

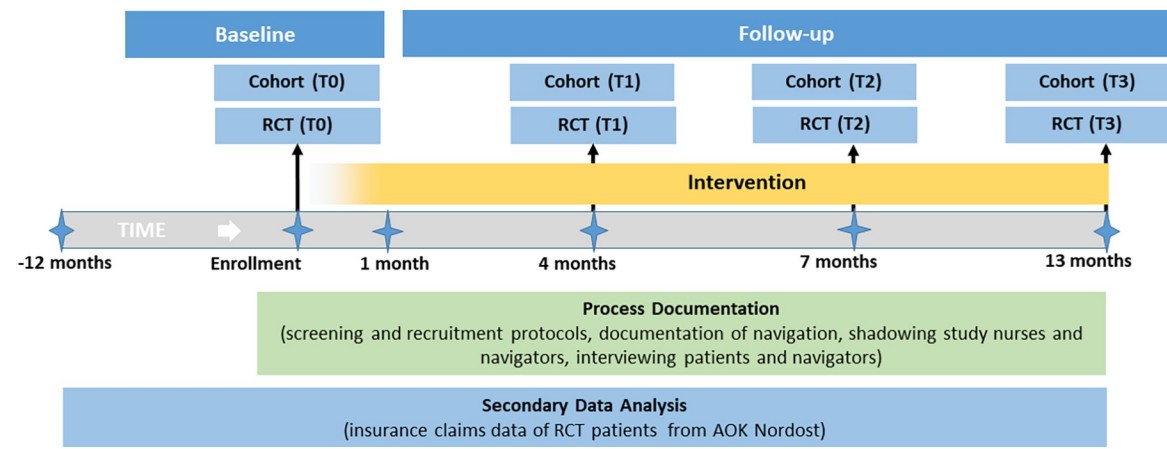

**Fig 2. Study design.**

- Caregiver of a patient (only for RCT) with any of the above diagnoses (with the patient's consent or existing legal representation).

- Age: ≥18 years

- Residence in Berlin or Brandenburg

- Insurance status with the AOK Nordost (for claims data analysis within RCT only)

  Exclusion Criteria are:

- Nursing home residence at time of study enrolment

- Patients who are not capable of informed consent and have no existing legal care

- Dementia (inclusion of caregivers is possible for the RCT)

- Language barrier (inclusion of caregivers is possible for the RCT)

## Recruitment and consent process

Eligible patients are screened according to inclusion and exclusion criteria in cooperation with the participating recruitment sites. Study information materials are distributed on the wards and patients or caregivers are actively approached by the study personnel. For patients and their caregivers not interested in participating in the RCT, participation in the parallel cohort is offered. RCT-participants, who are insured in the participating statutory health insurance company (AOK Nordost), are approached and informed about the secondary data analysis. The possibility for later contact and enrolment is provided to interested patients and caregivers. Those interested in participating are provided with the detailed study information. Patients and caregivers willing to participate give written informed consent. Study refusers are asked to complete a brief refuser questionnaire.

Arm 1: Navigation arm: Participants receive support of a personal patient navigator for one year.

Arm 2: Control arm: Participants receive a print brochure with regional support offers for patients with stroke/lung cancer and their caregivers.

Cohort: Eligible participants who do not want to be participate in the RCT for various reasons. They receive no patient navigation or brochure.
Recruitment period started in June 2021 until the end of August 2022.

## Intervention

**Intervention.**    Participants randomized to the intervention arm receive support of a personal patient navigator for up to one year with a minimum contact frequency every three months. The PN is individually tailored to the needs of participants and provides support in the domains i) provision of information for support for everyday life and further treatment options, such as counselling services, psychosocial and other support options, ii) help with application formalities, such as applying for a disability certificate or care degree and rehabilitation measures, iii), if necessary, establishing contact with (specialist) doctors, therapists and other support and care facilities.

**Control.**    Participants in the control arm receive a print brochure with regional support offers for patients with stroke/lung cancer and their caregivers, which has previously been developed [17].

**Cohort study.** Eligible patients who do not want to participate in the RCT are invited to take part in an observational cohort study, where they do not receive support by a personal navigator or a brochure. Cohort participants received the same questionnaires as RCT participants at the same time points. The cohort studies allow the comparison of health care needs and patient characteristics between participants with and without a subjective need for PN over time.

## Allocation to study arm within RCTs

Randomisation within each patient / participant group (stroke / lung cancer) is stratified by centre (Berlin, Brandenburg) and by type of participant (patient or caregiver). Randomisation is conducted by an independent biostatistician using the R Package randomizeR [21], after patients have completed the baseline assessment. Group assignment is sent to the study personnel via e-mail for each included patient.

## Outcomes

**Primary outcome.** The primary outcome is the feasibility of the intervention. We defined the primary feasibility criteria as: (1) At least 70% of patients randomized to the intervention arm have received at least one initial in-person navigator session, and (2) The dropout rate from the intervention arm of the RCT is less than 40% within one year.

In case these feasibility criteria are met, we will test the efficacy outcome 'satisfaction with care' as a further primary outcome (single item 'Overall, how satisfied are you with the general medical care?' with 5-point likert response scale) at the final follow-up (13 months after baseline, 12 month after the start of the intervention).

**Secondary outcomes.** Several feasibility and efficacy outcomes have been selected as secondary outcomes. Table 1 provides an overview of the selected feasibility indicators.

Secondary efficacy outcomes are several patient and caregiver reported outcomes (for full list of outcomes and used measures see Tables 2 and 3). Outcomes from the secondary data

**Table 1. Feasibility indicators.**

| Area of feasibility | Indicator |
| --- | --- |
| **Acceptance** | • % eligible patients interested to receive the navigator intervention |
| | • % consenting to the RCTs |
| | • % interested but unwilling to be randomized |
| | • % patients randomized and starting the intervention |
| | • **% patients continuing the intervention until the end (after 12 months)**\* |
| | • % patients adhering to the appointments with navigator |
| | • % patients and % navigators ,very satisfied' or ,satisfied' with the intervention |
| **Demand** | • Support needs |
| | • Number of navigator contacts |
| | • Delivery mode of the navigator interaction (in-person, phone, email) |
| | • Type of navigator services performed |
| **Implementation and Practicality** | • % of target population (potentially eligible patients) approached and informed about study at recruitment centres before discharge |
| | • **% enrolled patients who receive first in-person navigator session within scheduled timeframe**\* |
| | • % patient-navigator appointments carried out as planned |
| | • % of individual patient goals achieved |

\* Primary feasibility outcomes.

**Table 2. Measures for patients.**

| Item name | Number of items | Stroke | | | | Notes | Lung cancer | | | | Notes | Reference |
|---|---|---|---|---|---|---|---|---|---|---|---|---|
| | | B | T1 | T2 | T3 | | B | T1 | T2 | T3 | | |
| Present situation | 2 | X | - | - | - | | X | - | - | - | | self-developed |
| Age | 1 | X | - | - | - | | X | - | - | - | | [22] |
| Gender | 1 | X | - | - | - | | X | - | - | - | | self-developed |
| Health-related quality of life (generic) | 5 | X | X | X | X | | X | X | X | X | | [23–26] |
| Health-related quality of life (generic, scale) | 1 | X | X | X | X | | X | X | X | X | | [23–26] |
| Health-related quality of life (specific, symptoms) | 14 | - | - | - | - | specific for lung cancer | X | X | X | X | | [27] |
| Barthel-Index | 10 | X* | X | X | X | *medical charts | - | - | - | - | specific for stroke | [28] |
| Mental health | 4 | X | X | X | X | | X | X | X | X | | [29] |
| Distress (scale) | 1 | X | - | X | X | | X | - | X | X | | [30] |
| Utilization of medical care | 1 | X | X | X | X | | X | X | X | X | | Adapted from [31] |
| Utilization of inpatient care | 1 | X | X* | X* | X* | *+ period of time + clinic/infirmary | X | X* | X* | X* | *+ period of time + clinic/infirmary | Adapted from [31] |
| Utilization of emergency department | 2 | X | X | X | X | | X | X | X | X | | [22] |
| Utilization of therapy / counselling | 1 | - | X | X | X | | X | X | X | X | | Adapted from [31] |
| Utilization of rehabilitation | 1 | - | X | X | X | | - | X | X | X | | Adapted from [31] |
| Delayed care | 2 | X | X | X | X | | X | X | X | X | | [22] |
| Forgone Care (renunciation) | 2 | - | X | - | - | | X | X | X | - | | [32] |
| Satisfaction with care | 2 | X | X | X | X | | X | X | X | X | | self-developed |
| Information needs | 4 | X | X | - | - | | X | X | - | - | | [33] |
| Health literacy | 4 | - | X | - | X | | X | X | - | X | | [34] |
| Health literacy | 3 | X | X | - | X | | X | X | - | X | | [35] |
| Trust in medical care | 1 | X | X | X | X | | X | X | X | X | | [32] |
| Support needs | 15 | X | X* | X** | X** | *19 items ** 23 items | X* | X** | X*** | X*** | *16 items **20 items *** 23 items | [36] and self-developed items |
| Social support | 6 | X | - | - | - | | X | - | X | - | | [37] |
| People who support | 1 | X | X* | X* | X* | *+ category „emotional support" | X | X* | X* | X* | *+ category „emotional support" | Adapted from [31] |
| Loneliness | 6 | X | - | - | - | | X | - | - | - | | [38] |
| Social relationships | 9 | - | X | X | - | | - | X | X | X | | [39] |
| Self-efficacy | 7 | - | - | X | X | | X | X | X | X | | [40] |
| Smoking | 1 | - | X | X | X | | - | X | X | X | | [22] |
| Daily activity | 2 | - | X | X | X | | - | X | X | X | | [31] |
| Sleep | 2 | - | X | X | X | | - | X | X | X | | [31] |
| Use of computers | 2 | - | - | X | - | | - | - | X | - | | self-developed |
| Marital status | 2 | X | - | - | - | | X | - | - | - | | [22] |
| Living situation | 1 | X | X | X | X | | X | X | X | X | | self-developed |
| Pet | 1 | X | - | - | - | | X | - | - | - | | self-developed |
| Education | 2 | X | - | - | - | | X | - | - | - | | Adapted from [22] |
| Employment | 1 | X | X* | X* | X* | *+ vocational rehabilitation | X | - | X* | X* | *+ vocational rehabilitation | Adapted from [22] |
| Care level | 3 | X | X | X | X | | X | X | X | X | | Adapted from [31] |
| Medication adherence | 1 | - | - | - | X | | - | - | - | X | | Translated from [41] |
| Impairment / disability | 2 | - | X | X* | X* | *only disability | - | X* | X* | X* | *only disability | [22] |

(*Continued*)

**Table 2.** (Continued)

| Item name | Number of items | Stroke | | | | Notes | Lung cancer | | | | Notes | Reference |
|---|---|---|---|---|---|---|---|---|---|---|---|---|
| Chronic diseases | 1 | - | - | X | X | | - | - | X | X | | Adapted from [22] |
| Height / weight | 2 | X | - | - | - | | X | - | - | - | | Adapted from [22] |
| Nationality and migration status | 4 | X | - | - | - | | X | - | - | - | | Adapted from [22] |
| Restriction of contacts | 1 | - | X | X | X | | - | X | X | X | | [31] |
| Status of received care | 11 | X* | - | - | - | *only patients at home | - | - | - | - | | self-developed |
| Satisfaction with navigation | 13 | - | X | X* | X** | only in navigation group *new wording, +2 items, -1 item, -open questions **+1 item | - | X | X* | X** | only in navigation group *new wording, +2 items, -1 item, -open questions **+1 item | Adapted from [42] and self-developed |
| Satisfaction with brochure | 14 | - | - | - | X* | *only control group | - | - | - | X* | *only control group | self-developed |

analysis of insurance claims include utilization of medical care and costs, clinical outcomes such as mortality and cost-effectiveness (Table 4).

## Sample size and statistical testing

**RCT for stroke patients and their caregivers.** Based on assumptions about patient recruitment numbers that we would be able to achieve across a 1-year recruitment period at the study centres we made the following power considerations. It was estimated that 60–70% of patients may be reached for active recruitment after screening. Of these patients, we estimated a recruitment rate of 30–50%, depending on recruitment centre.

Based upon these considerations, we estimated a total enrolment of 460 stroke patients at the respective study sites. If 230 (50%) of the patients are randomised to the intervention arm, we assume that at least 95% (n = 219) will still have survived to receive PN at 4 weeks. For the primary feasibility outcomes this would result in the following: if 166 (75.8%) or more of these 219 patients receive the initial navigation session, the first feasibility criterion is met, as the 95% confidence interval of this proportion will not be less than 70% (95%CI: 70.1%-81.5%). We also assume that of these 230 patients (intervention arm), 85% (n = 196) survived after one year. If of these 196 patients, 65 (33.2%) or less are lost-to-follow-up, the second feasibility criterion will be met, as the 95% confidence interval of this proportion will be less than 40% (95% CI: 26.6%-39.8%).

The feasibility of the study is considered successful if both criteria are achieved.

In case of successful demonstration of feasibility, we additionally test the efficacy of the intervention in the study at the two-sided significance level $\alpha$ = 0.05 (hierarchical testing). All other outcomes are analysed in a secondary exploratory manner.

**RCT in lung cancer patients and their caregivers.** If 120 lung cancer patients are included in the study, and 60 (50%) of the patients are randomised to the intervention arm, we assume that at least 95% (n = 57) will still have survived to receive PN at 4 weeks. If 46 (80%) or more of these 57 patients receive the initial navigation session, the first feasibility criterion is met, as the 95% confidence interval of this proportion will not be less than 70% (95%CI: 70.5%-90.9%). We also assume that of these 57 patients (intervention arm), 75% (n = 43) survived after one year. If of these 43 patients, 11 (25.6%) or less are lost-to-follow-up, the second feasibility criterion is met, as the 95% confidence interval of this proportion will be less than 40% (95% CI: 12.5%-38.6%).

**Table 3. Measures for caregivers.**

| Item name | Number of items | Stroke | | | | Notes | Lung cancer | | | | Notes | Reference |
|---|---|---|---|---|---|---|---|---|---|---|---|---|
| | | B | T1 | T2 | T3 | | B | T1 | T2 | T3 | | |
| **Part 1: Information about patient (proxy rating)** | | | | | | | | | | | | |
| Barthel-Index | 10 | - | X | X | X | | - | - | - | - | Specific for stroke | [28] |
| Health-related quality of life (specific, symptoms) | 14 | - | - | - | - | Specific for lung cancer | - | X | X | X | | [27] |
| Health-related quality of life (generic, scale) | 1 | - | - | - | - | | - | X | X | X | | [23–26] |
| Pesent situation | 2 | X | - | - | - | | X | - | - | - | | self-developed |
| Age | 1 | X | - | - | - | | X | - | - | - | | [22] |
| Gender | 1 | X | - | - | - | | X | - | - | - | | self-developed |
| Utilization of medical care | 1 | X | X | X | X | | X | X | X | X | | [31] |
| Utilization of inpatient care | 1 | X | X* | X* | X* | *+ period of time + clinic/infirmary | X | X* | X* | X* | *+ period of time + clinic/infirmary | [31] |
| Utilization of emergency department | 2 | X | X | X | X | | X | X | X | X | | [22] |
| Utilization of therapy / counselling | 1 | - | X | X | X | | - | X | X | X | | [31] |
| Utilization of rehabilitation | 1 | - | X | X | X | | - | X | X | X | | Adapted from [31] |
| Delayed care | 2 | X | X | X | X | | X | X | X | X | | [22] |
| Forgone Care (renunciation) | 2 | - | X | - | - | | - | X | - | - | | [32] |
| Marital status | 2 | X | - | - | - | | X | - | - | - | | [22] |
| Legal guardianship | 2 | - | X | X | X | | - | X | X | X | | self-developed |
| Does relative live in household? | 1 | - | - | X | X | | - | X | X | X | | self-developed |
| Living situation | 1 | X | X | X | X | | X | X | X | X | | self-developed |
| Pet | 1 | X | - | - | - | | X | - | - | - | | self-developed |
| Education | 2 | X | - | - | - | | X | - | - | - | | Adapted from [22] |
| Employment | 1 | X | - | X* | X* | *+ vocational rehabilitation | X | - | X* | X* | *+ vocational rehabilitation | Adapted from [22] |
| Persons in household | 1 | - | X | X | X | | - | X | X | X | | [31] |
| Care level | 3 | X | X | X | X | | X | X | X | X | | [31] |
| Impairment / disability | 2 | - | X | X | X* | *only disability | - | X* | X* | X* | *only disability | [22] |
| Height / weight | 2 | X | - | - | - | | - | - | - | - | | Adapted from [22] |
| Nationality and migration status | 4 | X | - | - | - | | X | - | - | - | | [22] |
| **Part two: Information about caregiver (self-report)** | | | | | | | | | | | | |
| Age | 1 | X | - | - | - | | X | - | - | - | | [22] |
| Gender | 1 | X | - | - | - | | X | - | - | - | | self-developed |
| Relationship with the patient | 1 | X | - | - | - | | X | - | - | - | | self- developed |
| Health-related quality of life (generic) | 5 | X | X | X | X | | X | X | X | X | | [23–26] |
| Health-related quality of life (generic, scale) | 1 | X | X | X | X | | X | X | X | X | | [23–26] |
| Mental health | 4 | X | X | X | X | | X | X | X | X | | [29] |
| Distress | 1 | X | X | X | X | | X | X | X | X | | [30] |
| Satisfaction with care | 2 | X | X | X | X | | X | X | X | X | | self-developed |
| Information needs | 4 | X | X | - | - | | X | X | - | - | | [33] |
| Health literacy | 4 | X | X | - | X | | X | X | - | X | | [34] |
| Health literacy | 3 | X | X | - | X | | X | X | - | X | | [35] |

(*Continued*)

**Table 3.** (Continued)

| Item name | Number of items | Stroke | | | | Notes | Lung cancer | | | | Notes | Reference |
|---|---|---|---|---|---|---|---|---|---|---|---|---|
| | | B | T1 | T2 | T3 | | B | T1 | T2 | T3 | | |
| Trust in medical care | 1 | X | X | X | X | | X | X | X | X | | [32] |
| Support needs (in relation to patient) | 15 | X | X* | X** | X** | *13 items ** 18 items | X* | X** | X*** | X*** | *16 items **13 items *** 18 items | [36] |
| Social support | 6 | X | - | - | - | | X | - | - | - | | [37] |
| People who support (in relation to self) | 1 | X | X* | X* | X* | *+ category „emotional support" | X | X* | X* | X* | *+ category „emotional support" | [31] |
| Loneliness | 6 | X | - | - | - | | X | - | - | - | | [38] |
| Marital status | 2 | X | - | - | - | | X | - | - | - | | [22] |
| Living situation | 1 | X | X | X | X | | X | X | X | X | | self-developed |
| Use of computers | 2 | - | - | X | - | | - | | X | - | | self-developed |
| Pet | 1 | X | - | - | - | | X | - | - | - | | self-developed |
| Education | 2 | X | - | - | - | | X | - | - | - | | [22] |
| Employment | 1 | X | X | X | X | | X | X | X | X | | [22] |
| Care level | 3 | X | X | X | X | | X | X | X | X | | [31] |
| Care for patient | 1 | X | - | - | - | | X | - | - | - | | self-developed |
| Chronic diseases | 1 | X | - | - | X | | X | - | - | X | | [22] |
| Nationality and migration status | 4 | X | - | - | - | | X | - | - | - | | [22] |
| Satisfaction with navigation | 14 | - | X | X | X* | only navigation group * - 1 open question, +1 item | - | X | X | X* | only navigation group *new wording, +1 item | Adapted from [42] and self-developed |
| Satisfaction with brochure | 14 | - | - | - | X* | *only control group | - | - | - | X* | *only control group | self-developed |

The feasibility of the study is considered successful if both criteria are achieved and efficacy outcomes are analysed as described above.

## Data collection

**Process documentation (screening, recruitment, navigation).** Screening and recruitment are documented on a daily basis (number of patients screened, in-/exclusion criteria, refusal rates). Study nurses collect reasons for non-participation from patients who are eligible but refuse study participation. At recruitment hospitals, anonymous information on age, gender, disease severity and comorbidities from the entire cohort of non-participating patients who were treated at the hospital units during the recruitment period will be extracted from the

**Table 4. Outcomes of secondary data analysis.**

**Outcome**

- Utilization of medical care and costs
  - Outpatient and inpatient medical care
  - Rehabilitation
  - Remedies, e.g. physiotherapy
  - Medication
  - Nursing care
- Cost effectiveness (based on quality-adjusted life years (QALYs))
- Clinical Outcomes
  - (Time to) Rehospitalization
  - 1 year mortality/survival
  - Recurrent stroke event

hospital documentation system at study end, so that participants and non-participants can be compared on these characteristics.

The feasibility indicators (Table 1) are measured through detailed documentation throughout the recruitment and intervention process. Documentation files for screening and recruitment are kept separate from files used to document the PN.

Detailed assessment of medical data of participating patients is collected from medical charts.

**Patient reported outcomes.**    At baseline and prior to randomization, participants complete a questionnaire and at each follow-up, questionnaires are mailed to participants.

Reported hospitalizations during the follow-ups are verified by hospital records; an active search will be performed for hospitalizations in the clinic database of the recruiting hospitals. Study personnel is blinded to participants' group status. Vital status will be verified through the registration offices at completion of the follow-ups.

Tables 2 and 3 provide an overviews of the patient reported outcomes.

**Collection of insurance claims data.**    For the subsample of study patients, who are insured in the participating statutory health insurance company (AOK Nordost), the claims data during the study participation and one year before patients' study enrolment will be provided pseudonymized to the researcher after patients' consent. As the health insurance has a regional market share of 28% [43], we would expect such data to become available for about a quarter of all patients of the RCT. The data provided cover all in- and outpatients visits and treatments (including re-hospitalizations/recurrent stroke), data on medication, nursing care, rehabilitation, mortality, palliative care, healthcare costs, and the use of remedies and medical products. These data will allow to analyse the possible impact of the PN on the health care utilisation and will additionally used for conducting analysis on costs. For cost-effectiveness analyses, the results on mean health care utilization costs will be combined with primarily collected patient reported outcomes, such as quality of life.

**Participant observations and qualitative interviews.**    The associated qualitative study focuses on the delivery of the navigation program and the recruitment into the RCTs and cohort studies. The aim is to identify factors of the delivery and recruitment processes that are associated with success and failure of the feasibility.

To evaluate the recruitment strategy, we observe the processes at the study sites by shadowing the study nurses. Structured field notes and protocols are recorded of the observation. Furthermore, we conduct semi-structured interviews with our study nurses in regular intervals to capture their experiences with recruitment.

To assess the feasibility of our PN program from the deliverers´ perspective, we use participant observation of the weekly navigator team meetings as well as qualitative interviewing of the navigators in regular intervals throughout the intervention phase. A comprehensive documentation of navigation activities provides detailed information about patients' use of the navigation program.

The patients' perspective is assessed from qualitative interviews with patients (n = 30) enrolled in the RCT or cohort study. Selection of patients for the qualitative interviews is based on age, gender, and diversity in patients' use of navigation services and satisfaction with the navigator's work.

## Data analyses

**RCTs.** Feasibility criteria for both RCTs will be tested using 95%CI of proportions. The lower limit of the 95%CI of the proportion of patients in the intervention arm in each RCT who participated in at least one initial face-to-face navigator session should be above 70%, and the

higher limit of the 95%CI of the dropout rate in each RCT is below 40%. If both criteria are met and feasibility is therefore demonstrated, efficacy of both interventions will be tested separately using an ordinal regression model with 'satisfaction with care' (measured on a five point likert scale) 12 months after start of intervention as ordinal dependent variable, adjusted for baseline quality of life, centre, and for type of participant (patient or caregiver). If feasibility cannot be demonstrated, all other outcomes will be analysed in an exploratory manner. All secondary outcomes will be tested similarly using ANCOVA regression models adjusted for the particular baseline measure, centre, and type of participant [44]. Predefined sub-group analyses will be conducted to identify possible differences in intervention effects by calculating interaction effects and marginal effects.

All primary questions will be analysed confirmatory in the "full analysis set". In the case of missing values and assuming "missing at random" (MAR) or "missing completely at random" (MCAR), multiple imputation models are used to estimate the missing values.

**Cohort studies.**   In the parallel cohort studies the same outcomes will be analyse exploratory to compare patient reported outcomes and patient needs to those of participants who opted to enrol in the RCT.

**Economic analyses.**   To date, the availability of high quality economic analyses with regard to PN in stroke und lung cancer patients is scarce [45,46]. For that reason, the present study will be supplemented by a health economic evaluation. The health economic analyses will consist of a (1) cost comparison analysis, supplemented with a (2) cost-effectiveness analysis and will be part of the planned RCTs. The cost comparison analysis will be based on the health claims data collected for patient being member of the participating health insurance company. These data will provide information on kind and frequency of health resource consumption covering care of all sectors as well as data on cost reimbursement. Therefore, the planned cost analysis will cover the health insurance perspective. The cost results will be tested for relevant and statistically significant differences between the treatment groups. For measuring the cost-effectiveness of the intervention, the results of the cost comparison analyses will be combined with results of the self-rated quality of life and survival time during the study in terms of quality adjusted life years. QALYs will be calculated based on the results of the EQ-5D questionnaire [23–26] and mortality data identified during the observation period. In case of the superiority of the intervention in terms of QALYs and increased costs for the intervention group, the incremental costs per QALY gained (ICER) will be obtained by combining cost- and effect differences between treatment groups. In case of increased costs and improved quality of life, the ICER indicates the cost-effectiveness by adopting an internationally accepted threshold (e.g., $\leq$ 50,000 Euro per QALY gained). All cost-effectiveness results will also take into account subgroups (e.g. by gender) and sensitivity analyses and finally be visualised in a cost-effectiveness plane.

**Qualitative study.**   All qualitative data will be analysed thematically with regards to the different research questions and outcomes [47]. Interviews with study nurses, navigators and patients will be audio-recorded and transcribed verbatim. To ensure the quality of analysis, it will be conducted within the research group and regularly discussed in qualitative research forums.

## Data management

Data is gained from several data sources and collected in different databases.

Data from participants is pseudonymized and transferred from paper questionnaires and managed using REDCap electronic data capture tools hosted at Charité Universitätsmedizin Berlin [48,49].

AOK Nordost will provide the claims data via a secure exchange platform using a matching pseudonymization. Data provision will be overseen and permitted by the responsible supervisory authority. Furthermore, a detailed screening log is documented for detailed process evaluation and collection of data (age, sex, diagnosis, comorbidities) for the comparison of participants and non-participants. For non-participants, this data is analysed completely anonymized.

Navigators document their activities in a separate database (PN file).

A data safety plan was developed and approved by the data safety department of the Charité Universitätsmedizin Berlin.

The coordinator of the NAVICARE network will conduct one independent audit during the recruitment phase of the study period. The coordinator of NAVICARE is independent of the CoreNAVI research team.

## Ethics statements

This protocol follows the standards of the Helsinki Convention of good clinical practices. The Ethics Committee of the Charité–Universitätsmedizin Berlin (EA2/249/20) and Medizinische Hochschule Brandenburg–Theodor Fontane (Z-01-20210517) have approved the project.

## Availability of data and material

Individual participant data from quantitative assessments that underlie the results reported in published articles after de-identification (text, tables, figures and appendices) will be made available 12 months after publication to researchers who provide a methodologically sound proposal (e.g., for systematic reviews, meta-analyses, individual participant data meta-analyses). Study center individual patient data will not be disclosed because of the possibility of re-identification of patients.

## Discussion

Here, we describe a study that aims to evaluate the feasibility of a PN programme that was developed using an empirical approach including multiple data sources and perspectives [19]. A key feature of the PN intervention is the focus on individual patients' needs and preferences in all aspects and questions that arise along their care path for their health-related wellbeing. We chose two prototypic age-associated diseases, lung cancer and stroke, that are characterized in very different disease manifestation and trajectories. This way we will gain insights about tasks of the navigation service that are shared between diseases and the ones specific to a disease for a feasible navigation model for age-associated diseases. The study was designed as a feasibility study as we deem it essential for an effective evaluation of a programme's efficacy to first address questions on the programme's acceptance by patients, caregivers and health care providers, their evaluation of the navigation services provided, the demand for such a programme and the practicality in the real-life clinical environment.

For study participation, patients are actively approached in hospital wards and outpatient hospital clinics. Despite the challenges of approaching patients in a hospital setting, we chose this approach in order to not lose track of vulnerable patients in the gap between hospital and ambulatory care. Using an open recruitment strategy by primarily actively recruiting in the recruitment centre, but additionally providing information material to hospitals and other care institutions involved in the care of the respective indication, we intend to broaden the access for potentially interested and eligible patients and caregivers.

### Covid considerations

We adapted the recruitment strategy by adding an outpatient rehabilitation centre in Berlin to the recruiting sites, where advertising material is distributed to eligible patients. In addition, advertising material is distributed through ambulatory healthcare providers, and at the Berlin sites, eligible stroke patients received an invitation letter from the clinic during times where study nurses had no access to the hospital wards.

Navigators intensified networking with healthcare institutions in the ambulatory sector. In the PN programme intervention, in-person meetings temporarily had to be reduced to a minimum and a digital tool for virtual meetings with navigators was implemented.

### Dissemination

Besides publications in peer reviewed journals, findings will be disseminated to patients and family caregivers, researchers, healthcare providers and other stakeholders as well as the general public through the NAVICARE network's website (https://navicare.berlin/de/), twitter account (https://twitter.com/NAVICARE_Berlin), annual symposia, presentations at partnering institutions' events and scientific conferences, and peer-reviewed publications. The study has been registered at the German Clinical Trials register prior start of recruitment (DRKS00025476).

## Supporting information

**S1 Checklist. SPIRIT 2013 Checklist: Recommended items to address in a clinical trial protocol and related documents\*.**
(PDF)

**S1 File.**
(PDF)

**S2 File.**
(PDF)

## Acknowledgments

We wish to thank Sandra Wendlandt for her support during the writing process of the paper.

## Author Contributions

**Conceptualization:** Kathrin Gödde, Hella Fügemann, Ute Goerling, Nina Rieckmann, Christine Holmberg.

**Funding acquisition:** Ute Goerling, Ulrike Grittner, Andreas Meisel, Thomas Reinhold, Susanne Schnitzer, Nina Rieckmann, Christine Holmberg.

**Methodology:** Kathrin Gödde, Hella Fügemann, Ute Goerling, Ulrike Grittner, Raphael Kohl, Andreas Meisel, Thomas Reinhold, Susanne Schnitzer, Nina Rieckmann, Christine Holmberg.

**Project administration:** Kathrin Gödde, Hella Fügemann, Nina Rieckmann, Christine Holmberg.

**Resources:** Ute Goerling, Andreas Meisel, P. Markus Deckert, Nikolaj Frost, Stephan J. Schreiber.

**Supervision:** Nina Rieckmann, Christine Holmberg.

**Writing – original draft:** Kathrin Gödde, Nina Rieckmann, Christine Holmberg.

**Writing – review & editing:** Hella Fügemann, Ute Goerling, Ulrike Grittner, Raphael Kohl, Andreas Meisel, Thomas Reinhold, Susanne Schnitzer, P. Markus Deckert, Nikolaj Frost, Stephan J. Schreiber.

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
