## [Decision Letter · Decision Letter 0]

30 Jan 2023

PONE-D-22-19723Feasibility of a patient-oriented navigation programme for patients with Lung Cancer or Stroke in Germany: protocol of the CoreNAVI studyPLOS ONE

Dear Dr. Gödde,

Thank you for submitting your manuscript to PLOS ONE. After careful consideration, we feel that it has merit but does not fully meet PLOS ONE’s publication criteria as it currently stands. Therefore, we invite you to submit a revised version of the manuscript that addresses the points raised during the review process.

We look forward to receiving your revised manuscript.

Kind regards,

Walid Kamal Abdelbasset, Ph.D.

Academic Editor

PLOS ONE

Journal Requirements:

Reviewers' comments:

Reviewer's Responses to Questions

**Comments to the Author**

1. Does the manuscript provide a valid rationale for the proposed study, with clearly identified and justified research questions?

Reviewer #1: Yes

Reviewer #2: Yes

Reviewer #3: Yes

2. Is the protocol technically sound and planned in a manner that will lead to a meaningful outcome and allow testing the stated hypotheses?

Reviewer #1: Yes

Reviewer #2: Yes

Reviewer #3: Yes

3. Is the methodology feasible and described in sufficient detail to allow the work to be replicable?

Reviewer #1: Yes

Reviewer #2: Yes

Reviewer #3: Yes

4. Have the authors described where all data underlying the findings will be made available when the study is complete?

Reviewer #1: Yes

Reviewer #2: Yes

Reviewer #3: No

5. Is the manuscript presented in an intelligible fashion and written in standard English?

Reviewer #1: Yes

Reviewer #2: Yes

Reviewer #3: Yes

6. Review Comments to the Author

You may also provide optional suggestions and comments to authors that they might find helpful in planning their study.

Reviewer #1: Dear Editor

Plos One

Thank you very much for giving me the opportunity to review the manuscript entitled (Feasibility of a patient-oriented navigation programme for patients with Lung Cancer or Stroke in Germany: protocol of the CoreNAVI study)

The study idea is good. I have some comments to improve the manuscript.

The abstract and introduction sections are so long. The authors should review these sections and remove unnecessary words/sentences.

Reviewer #2: The authors have designed this protocol to show the Feasibility of a patient-oriented navigation program for patients with Lung Cancer or Stroke in Germany. It is a well-designed and well protocol

some points should be explained .

Line 300 301 302

Why did you select these percent (above 70 % and below 40 %)

Line 304 using an ordinal regression model

Explain the selection for that model

Line 307 308

Discuss the selection of ANCOVA for Your Study

Reviewer #3: The research design seems to be conducted in a proper way, The research idea is innovative and focused a new era. Only one explanation need to be clarified about participants age why didn't you make an upper limit for participants age to avoidphysiological cognitive changes?

7. PLOS authors have the option to publish the peer review history of their article (what does this mean?). If published, this will include your full peer review and any attached files.

Reviewer #1: **Yes: **Sherif Mohamed

Reviewer #2: No

Reviewer #3: No

---

## [Author Response · Author response to Decision Letter 0]

16 Mar 2023

Answer to the Editor and Reviewers’ comments (this response has also been attached as pdf file to this submission):

Dear Editor,

Thank you very much for the opportunity to submit a revised manuscript for our work entitled ‘Feasibility of a patient-oriented navigation programme for patients with Lung Cancer or Stroke in Germany: protocol of the CoreNAVI study’. We now submitted a marked-up copy as well as an unmarked version of the revised manuscript via the PLOS One submission system.

We are thankful for the helpful comments of the reviewers and are happy to address the raised questions (please see below).

Besides the points described below in the response to the reviewers, the following changes have been made to the manuscript:

Page 4, line 136: correction of wording to ‘We further aim to provide estimates of efficacy regarding selected patient reported outcomes,…’

Page 4, line 142-143: correction and rewording for clarification to ‘Does a patient-oriented navigation programme impact estimates of efficacy (patient reported outcome: satisfaction with care)?’

Page 5, line 170: correction of sentence to ‘This is an open-labelled study since blinding was not possible.’

Page 10, line 288: rewording for clarification to ‘All other outcomes are analysed in a secondary exploratory manner.’

Page 17, line 473-474: addition of an Acknowledgement section

Page 20-25: Table 2 and Table 3: corrections in referencing and title

Page 26, Author’s contributions: Correction of abbreviations

Throughout manuscript: Correction of typos (e.g. page 5, line 180)

Response to reviewer’s comments:

Reviewer #1: Dear Editor Plos One

Thank you very much for giving me the opportunity to review the manuscript entitled (Feasibility of a patient-oriented navigation programme for patients with Lung Cancer or Stroke in Germany: protocol of the CoreNAVI study)

The study idea is good. I have some comments to improve the manuscript.

The abstract and introduction sections are so long. The authors should review these sections and remove unnecessary words/sentences.

Answer: We thank the reviewer for the helpful comment. We now revised the abstract and introduction part in order to shorten it by deleting unnecessary and redundant words and sentences.

See pages 2-4, abstract and introduction 

Reviewer #2: The authors have designed this protocol to show the Feasibility of a patient-oriented navigation program for patients with Lung Cancer or Stroke in Germany. It is a well-designed and well protocol. Some points should be explained .

Line 300 301 302

Why did you select these percent (above 70 % and below 40 %)

Answer: These percentages were defined in a discussion process between the participating researchers taking into account their own and published experiences. Criteria were selected with the aim to find indicators that appropriately balance the lost-to-follow up/dropout rate that can be expected in navigation intervention studies in severely affected patients on the one hand and defining a conclusive indicator of feasibility to reflect acceptance/demand/practicality of the intervention on the other hand. 

(see for comparable approaches for example: Forster, A., Hartley, S., Barnard, L. et al. An intervention to support stroke survivors and their carers in the longer term (LoTS2Care): study protocol for a cluster randomised controlled feasibility trial. Trials 19, 317 (2018). https://doi.org/10.1186/s13063-018-2669-5 or Walters K, Frost R, Kharicha K, Avgerinou C, Gardner B, Ricciardi F, Hunter R, Liljas A, Manthorpe J, Drennan V, Wood J, Goodman C, Jovicic A, Iliffe S. Home-based health promotion for older people with mild frailty: the HomeHealth intervention development and feasibility RCT. Health Technol Assess. 2017 Dec;21(73):1-128. doi: 10.3310/hta21730.)

We reworded the sentence ‘The programme is categorised as feasible if:’ to ‘We defined the primary feasibility criteria as:’ to indicate that definition of feasibility criteria was done by participating researchers.

See page 8, line 251-252

Line 304 using an ordinal regression model

Explain the selection for that model

Answer: We selected this model as the outcome “satisfaction with care” is measured on a five point Likert-scale and therefore an ordinal scaled measure. That is the reason why we would use an ordinal regression model here. Additionally, we want to adjust for centres, baseline quality of life and type of participant. To do so we use a multiple ordinal regression model.

To clarify this for the reader we now inserted the used scaling in the text.

See page 13, line 361 and 362: …, efficacy of both interventions will be tested separately using an ordinal regression model with ‘satisfaction with care’ (measured on a five point likert scale) 12 months after start of intervention as ordinal dependent variable, …

Line 307 308

Discuss the selection of ANCOVA for Your Study

Answer: We selected ANCOVA models as the state of the art regression models for testing group differences in randomized trials as they allow the adjustment for the specific baseline variables and adjustment for stratification variables, if stratified randomization was done as also described for the present study. See for example Senn, S. (2006), Change from baseline and analysis of covariance revisited. Statist. Med., 25: 4334-4344. https://doi.org/10.1002/sim.2682

We now cited this study to make the selection of the method more clear to the reader. See page 13, line 366

Reviewer #3: The research design seems to be conducted in a proper way, The research idea is innovative and focused a new era. Only one explanation need to be clarified about participants age why didn't you make an upper limit for participants age to avoid physiological cognitive changes?

Answer: We thank the reviewers for this important question. One of the aims of evaluating the feasibility of the patient navigation model was to investigate if the most vulnerable patients are accessed with the used outreach strategy. During the development process of the model, older age was defined as one of the vulnerability characteristics (see Gödde et al. Development of a patient-oriented navigation model for patients with lung cancer and stroke in Germany. BMC Health Serv Res. 2022;2 2(1):785.). Hence, we decided not to set an upper limit for recruitment here to be able to investigate the feasibility of the navigation programme for participants also of older age. However, patients were only enrolled in the study if capable for informed consent. Alternatively, caregivers were able to take part in the study if legal care existed.

---

## [Decision Letter · Decision Letter 1]

12 Jun 2023

Feasibility of a patient-oriented navigation programme for patients with Lung Cancer or Stroke in Germany: protocol of the CoreNAVI study

PONE-D-22-19723R1

Dear Dr. Gödde,

We’re pleased to inform you that your manuscript has been judged scientifically suitable for publication and will be formally accepted for publication once it meets all outstanding technical requirements.

Kind regards,

Uwe Konerding

Academic Editor

PLOS ONE

Additional Editor Comments (optional):

Reviewers' comments:

Reviewer's Responses to Questions

**Comments to the Author**

1. Does the manuscript provide a valid rationale for the proposed study, with clearly identified and justified research questions?

Reviewer #1: Yes

Reviewer #2: Yes

2. Is the protocol technically sound and planned in a manner that will lead to a meaningful outcome and allow testing the stated hypotheses?

Reviewer #1: Yes

Reviewer #2: Yes

3. Is the methodology feasible and described in sufficient detail to allow the work to be replicable?

Reviewer #1: Yes

Reviewer #2: Yes

4. Have the authors described where all data underlying the findings will be made available when the study is complete?

Reviewer #1: Yes

Reviewer #2: Yes

5. Is the manuscript presented in an intelligible fashion and written in standard English?

Reviewer #1: Yes

Reviewer #2: Yes

6. Review Comments to the Author

You may also provide optional suggestions and comments to authors that they might find helpful in planning their study.

Reviewer #1: Dear Authors

Thank you very much for doing your efforts to improve the manuscript. You covered the needed points in this revised manuscript. I think that the revised manuscript is ready for publication in its current form

Reviewer #2: thanks alot for your response

7. PLOS authors have the option to publish the peer review history of their article (what does this mean?). If published, this will include your full peer review and any attached files.

Reviewer #1: **Yes: **Sherif Mohamed

Reviewer #2: No

---

## [Editor Report · Acceptance letter]

19 Jun 2023

PONE-D-22-19723R1 

Feasibility of a patient-oriented navigation programme for patients with Lung Cancer or Stroke in Germany: protocol of the CoreNAVI study 

Dear Dr. Gödde:

I'm pleased to inform you that your manuscript has been deemed suitable for publication in PLOS ONE. Congratulations! Your manuscript is now with our production department. 

Kind regards, 

on behalf of

Dr. Uwe Konerding 

Academic Editor

PLOS ONE